# Comparative Mitogenomics in the Genus *Chlorophorus* (Coleoptera: Cerambycidae) and Its Phylogenetic Implications

**DOI:** 10.3390/insects16010008

**Published:** 2024-12-27

**Authors:** Zhengju Fu, Lu Chen, Lichao Tian, Zongqing Wang, Zhu Li

**Affiliations:** 1College of Plant Protection, Southwest University, Chongqing 400715, China; fuzhengju2024@163.com (Z.F.); chenlu120415@163.com (L.C.); 2Chongqing Landscape and Gardening Research Institute, Chongqing 401329, China; tian_lc1916@yeah.net; 3Chongqing Key Laboratory of Germplasm Innovation and Utilization of Native Plants, Chongqing 401329, China

**Keywords:** Clytini, *Chlorophorus*, mitochondrial genome, phylogeny, monophyly

## Abstract

The genus *Chlorophorus*, the second largest genus of Clytini, belongs to Coleoptera, Cerambycidae. However, phylogenetic relationships within the genus remain unclear and the classification of subgenera based on elytral marking patterns remains untested. The mitochondrial genome has been widely used to reveal insect phylogeny and evolution. In this study, the complete mitochondrial genomes of 21 *Chlorophorus* species were sequenced, assembled, and annotated. A detailed comparative analysis was conducted on mitochondrial features, including nucleotide composition, relative synonymous codon usage (RSCU), and the evolutionary rates of protein-coding genes. Phylogenetic analysis was performed using mitochondrial genome data from 35 species, which included 12 outgroup species. The findings support the monophyly of the genus *Chlorophorus* and provide insights into certain subgenera. Additionally, a comparative examination of male genitalia revealed that the sclerites of the endophallus might exhibit phylogenetic signals relevant to the genus. The results of this study enhance our understanding of taxonomy and phylogenetic relationships within the genus *Chlorophorus*.

## 1. Introduction

The family Cerambycidae, commonly referred to as longhorn beetles or longicorn beetles, is one of the most diverse families within the order Coleoptera, comprising over 30,000 known species worldwide, with numerous additional species yet to be described. The genus *Chlorophorus* Chevrolat, 1863, has traditionally been placed within the tribe Clytini. Recently, Zamoroka [1] proposed a new tribe, Chlorophorini, and a new subtribe, Chlorophorina, based on the type genus *Chlorophorus* Chevrolat, 1863. However, Chlorophorini was subsequently synonymized by Lazarev [2]. *Chlorophorus* species are predominantly distributed in the Oriental and Palearctic regions, with 18 and 8 taxa recorded in the Afrotropical and Australian regions, respectively. Notably, only one species, *C. annularis* (Fabricius, 1787), has been introduced to the Nearctic and Neotropical region [3].

Both adults and larvae of *Chlorophorus* are phytophagous, with some species—particularly *C. annularis*—recognized as significant agricultural and forestry pests. This species originally inhabits temperate to subtropical regions of Southeast Asia and poses threats not only to bamboo timber and products but also to a wide array of other plant families, including Aceraceae, Anacardiaceae, Betulaceae, Dipterocarpaceae, Hamamelidaceae, Juglandaceae, Lamiaceae, Malvaceae, Mimosaceae, Paeoniaceae, Poaceae, Rosaceae, Ulmaceae, and Vitaceae [4]. Additionally, the larvae of *C. diadema* (Motschulsky, 1854) and *C. caragana* Xie & Wang, 2012 are known to feed on the branches and trunks of plants, leading to significant weakening or death of the infested plants [5,6,7].

Currently, the genus *Chlorophorus* is recognized to comprise 302 species and subspecies worldwide [8]. Numerous taxonomists, including Pic [9,10,11,12,13,14,15,16,17], Gressitt [18,19], Gressitt and Rondon [20], Holzschuh [21,22,23,24,25,26,27,28,29,30,31,32], and Viktora [33,34,35,36,37], have significantly contributed to the taxonomic study of this genus. Notably, since 1990, the number of newly described taxa has rapidly increased in the last three decades, with more than 100 taxa described primarily from Laos, Thailand, Vietnam, Nepal, India, and China and so on. In light of this diversity, Özdikmen [38] conducted a study focusing on Turkish species and proposed a subgeneric arrangement comprising five subgenera. This was further expanded, with Özdikmen [3] evaluating 242 species based on elytral markings and classifying them into 36 subgenera, including 31 newly described subgenera.

Despite these advances, the monophyly of *Chlorophorus* remains unconfirmed, as previous studies indicate that the genus is polyphyletic. Lee and Lee [39] found that *C. diadema* forms a separate clade from other *Chlorophorus* species based on the six genes and only three *Chlorophorus* species included in their study. Similarly, Zamoroka [1] demonstrated that the genus *Chlorophorus* is completely polyphyletic based on the analysis of three mitochondrial genes and two nuclear genes. Based on these results, the genus *Sparganophorus* Zamoroka, 2021 was established using *Clytus diadema* Motschulsky, 1854, as a type species, and the statuses of *Humeromaculatus* Özdikmen, 2011, and *Perderomaculatus* Özdikmen, 2011, were elevated from subgenus to genus [1]. Additionally, the new subgenus of *Humeromaculatus* (*Viridiphorus*) Zamoroka, 2021, was established. Nevertheless, these classifications are not supported by Özdikmen [3] and Tavakilian and Chevillotte [8]. Furthermore, Lazarev [2] synonymized *Sparganophorus* Zamoroka, 2021, with *C.* (*Humeromaculatus)* Özdikmen, 2011, and *C.* (*Brevenotatus)* Özdikmen, 2022, with *C.* (*Viridiphorus*) Zamoroka, 2021.

The mitogenome has been widely used to study phylogeny in Coleoptera at various taxonomic levels [40,41,42,43,44,45,46,47,48,49]. Mitochondrial genomes provide more information compared to single molecular markers, facilitating a more comprehensive understanding of phylogenetic relationships. However, only two complete mitochondrial genomes of *Chlorophorus* have been recorded in the NCBI database (https://www.ncbi.nlm.nih.gov/, accessed on 9 October 2024).

In this study, mitochondrial genome sequences were obtained for 21 *Chlorophorus* species. The main mitochondrial genome characteristics, nucleotide composition, relative synonymous codon usage (RSCU), and evolutionary rates of protein-coding genes (PCGs) were comparatively analyzed across these *Chlorophorus* species, and their phylogenetic relationships were reconstructed.

## 2. Materials and Methods

### 2.1. Taxon Sampling

Twenty-one new mitochondrial genome sequences were obtained in this study. All the studied specimens were deposited in the Insect Collection, College of Plant Protection, Southwest University, Chongqing, and stored in 100% ethanol at −30 °C until use. The specimen information is shown in Table 1. The specimens were examined using a stereo microscope (Optec SZ780, Chongqing, China) and identified on the basis of morphological characteristics using the descriptive literature on each species and the type images [18,19,20,21,22,23,24,25,26,27,28,29,30,31,32,33,34,35,36,37,50]. Photographs were taken using a digital camera (Canon EOS7D, Tokyo, Japan and Helicon Focus 5.2, Helicon Soft Limited, Kharkov, Ukraine). For detailed examination, male genitalia were extracted from specimens, cleared in 10% NaOH. The male genitalia were imaged using a stereomicroscope (Leica M205A, Leica Microsystems AG, Wetzlar, Germany).

### 2.2. DNA Extraction, Mitogenome Sequencing, Assembly, Annotation, and Sequence Analyses

Genomic DNA was extracted from an adult’s muscle tissue of the prothorax and legs by the modified CTAB method [54]. The mitogenomes were sequenced via whole-genome shotgun sequencing at Personal Biotechnology Co., Ltd. Shanghai, China. The mitogenome sequencing was conducted on the Illumina Novaseq (150 bp paired-end reads and a 350 bp insert size) platform. Fastp [55] was used to remove low-quality reads. Approximately 3 Gb of clean data were finally obtained for each sample. Mitochondrial genome assembly was carried out using GetOrganelle v.1.7.5.0 with k-mer sizes of 21, 45, 65, 85, and 105 and a t-value of 15 [56] using the clean reads from sequencing.

Gene annotation was conducted using Geneious Prime 2024.0.4 [57]. The ribosomal RNA genes (rRNAs) and protein-coding genes (PCGs) were annotated and manually corrected in relation to the relevant species [51]. The secondary structures of 22 tRNAs were predicted using the MITOS Web Server [58] with the invertebrate genetic code and default parameters. The circular maps of the mitochondrial genomes were visualized using Geneious Prime.

The nucleotide composition bias was calculated using the formulas AT-skew = (A% − T%)/(A% + T%) and CG-skew = (G% − C%)/(G% + C%) [59]. The rate of non-synonymous (Ka) to synonymous (Ks) substitutions of 13 PCGs was calculated using DnaSP v.6.1.0 [60]. The relative synonymous codon usage (RSCU) of 13 protein-coding genes was determined using MEGA v.11 [61].

### 2.3. Phylogenetic Analyses

The phylogenetic relationships within the genus *Chlorophorus* were inferred from 35 mitogenome sequences, with 12 species of Cerambycinae and Prioninae used as outgroups. In this study, 13 PCGs, 2 rRNAs, and 22 tRNAs genes were extracted from the mitochondrial genome. The phylogenetic analyses were conducted based on 4 datasets: (1) 15 genes (13 PCGs and 2 rRNAs), (2) 37 genes (13 PCGs, 2 rRNAs, and 22 tRNAs), (3) PCG123 (including three codon positions of the 13 PCGs) and (4) PCG12 (including the 1st and 2nd codon positions of the 13 PCGs).

Sequence alignments were performed using MAFFT v7.475 [62,63]. The gaps and the ambiguous sites from multiple alignments were trimmed using Gblocks 0.91 [64,65] with default parameters. PhyloSuite v.1.2.3 [66] was used to separately concatenate the four datasets. We assessed heterogeneity and substitution saturation for each dataset using AliGROOVE v1.08 [67] and DNAMBE v5 [68]. ModelFinder [69] was used to evaluate the optimal partitioning strategy and evolutionary model for the datasets (Appendix A). The supermatrix is provided in the Appendix A.

We reconstructed the phylogeny of *Chlorophorus* based on maximum likelihood (ML) and Bayesian inference (BI) analyses. The BI analyses were conducted using MrBayes 3.2.6 [70] with four independent Markov chains, which were run for a total of 10 million generations. Samples were taken every 1000 generations, and the first 25% of the trees were discarded as burn-in. Convergence was assessed until the average standard deviation of split frequencies was less than 0.01. The maximum likelihood (ML) analyses were performed using IQ-TREE version 1.6.12 [71]. The support for each node was assessed based on 1000 bootstrap replicates. Additionally, the third and fourth datasets were analyzed under the default CAT + GTR model by using PhyloBayes MPI version 1.9 [72], running two Markov Chain Monte Carlo (MCMC) independently and the two runs had satisfactorily converged (maxdiff less than 0.1). The phylogenetic trees were then visualized using Interactive Tree of Life (iTOL) [73].

## 3. Results and Discussion

### 3.1. General Mitogenome Features

In this study, 21 complete mitochondrial genomes of *Chlorophorus* species were sequenced and submitted to GenBank, under the accession numbers PQ239536–PQ239556 (Table 1). The data for the newly sequenced mitochondrial genomes were combined with those of two *Chlorophorus* species (*C. annularis* and *C. diadema diadema*) from NCBI. We used a total of 21 *Chlorophorus* species for comparative analysis of the genome sequence length, nucleotide composition, relative synonymous codon usage, and evolutionary rates of protein-coding genes.

#### 3.1.1. Mitogenome Organization and Gene Content

All newly obtained sequences were characterized and found to contain a total of 37 genes, including 13 protein-coding genes (PCGs), 2 ribosomal RNAs (rRNAs), and 22 transfer RNAs (tRNAs) in addition to 1 non-coding region (control region). Of these, 9 PCGs and 14 tRNAs are encoded on the majority strand (J strand), while the remaining genes, including 4 PCGs, 8 tRNAs, and 2 rRNAs, are transcribed on the minority strand (N strand) (Appendix A). The gene arrangements of these 21 species appear highly conserved, showing no evidence of gene rearrangement and exhibiting a very compact organization similar to that of ancestral insect mitochondrial genomes [74].

The lengths of the 21 *Chlorophorus* mitochondrial genomes range from 15,387 bp (*C. diadema diadema*) to 15,779 bp (*C. douei*), with the variation primarily attributed to the length of the non-coding region. The lengths of these genes across these genomes range from 11058 bp to 11109 bp for the 13 PCGs, 1431 bp to 1449 bp for the 22 tRNAs, and 2015 bp to 2036 bp for the 2 rRNAs. The non-coding regions range from 813 bp (*C. diadema diadema*) to 1205 bp (*C. douei*) in length. Additionally, the spacer and overlapping regions were identified between some genes, ranging from 19 to 65 bp, among which the longest spacer region is found between *tRNA*-*Ser2* and *ND1*. Gene overlaps ranged from 1 to 7 bp in length.

#### 3.1.2. Nucleotide Composition

The AT contents of the 21 *Chlorophorus* species are shown in Appendix A. The A + T contents range from 63% to 76.8%, with the lowest found in *C. douei* and the highest in *C. lingnanensis*. The A + T contents in PCGs (69.06–72.84%), tRNAs (74.12–75.15%), rRNAs (73.51–75.32%), and CR (60.5–74.69%) exceed the G + C contents. Therefore, the base content of the mitochondrial genomes of the 21 species showed a significant A + T bias in the nucleotide compositions.

GC-skew and AT-skew are commonly used to indicate differences in mitochondrial genome base compositions. By analyzing the mitochondrial genome data of *Chlorophorus* species, we found that the AT-skew values are positive, ranging from 0.001 to 0.145, while the GC-skew values are negative, ranging from −0.333 to −0.135. This suggests an obvious bias toward the use of A and C throughout the entire genome of all analyzed *Chlorophorus* species (Appendix A). Overall, the nucleotide composition displayed features typical of Coleoptera mitogenomes [75,76].

#### 3.1.3. Protein-Coding Genes and Codon Usage

The *Chlorophorus* mitochondrial genome contains 13 protein-coding genes, of which 9 (*COI*, *COII*, *COIII*, *ATP6*, *ATP8*, *ND2*, *ND3*, *CYTB*, and *ND6*) are encoded on the J strand and 4 (*ND1*, *ND4*, *ND4L*, and *ND5*) are encoded on the N strand. The Ka/Ks values of 13 PCGs in the *Chlorophorus* mitogenomes are displayed in Figure 1, ranging from 0.01869 to 0.22667. From this, the evolution rates of the 13 PCGs are found to follow the order *ATP8* > *ND6* > *ND4L* > *ND5* > *ND3* > *ND4* > *ND2* > *ND1* > *CYTB* > *COII* > *COIII* > *ATP6* > *COI*, with the Ka/Ks values of *COI* being the lowest. This indicates that *COI* is the most highly conserved of the 13 PCGs and thus the slowest to evolve. Conversely, the *ATP8* gene exhibits the highest evolutionary rate.

Start and stop codons usage in the *Chlorophorus* protein-coding genes was statistically analyzed (Appendix A). Generally, most of the PCGs were found to use the conventional start codon ATN (N represents A, T, C, or G). The exceptions are *COI*, which uses AAC/AAT, and some *ND1*, which use TTG. All 13 PCGs use TAA/TAG or a single T as stop codons, and the T is replenished by post-transcriptional polyadenylation at the 3′ end [77], consistent with previous studies on Cerambycidae [75,76,78,79]. The statistics of the relative synonymous codon usage showed that the most commonly used amino acids in the *Chlorophorus* mitochondrial genome are Ile, Phe, and Leu1, which are also the most common amino acids in many other insect species [80,81,82,83]. Comparative analyses of the 21 samples indicate that codon use is conservative, with the majority exhibiting common utilization patterns. For instance, A and U are more commonly used than G and C, and the three most commonly used codons in 21 species of the genus *Chlorophorus* are AUU, UUU, and UUA (Appendix A).

#### 3.1.4. tRNA and rRNA Genes

The sizes of the 22 tRNAs range from 60 bp (*tRNA*-*Cys*) to 71 bp (*tRNA*-*Lys*). In terms of secondary structure, the tRNAs exhibit a classical cloverleaf structure with the conventional four arms, with the exception of *tRNA*-*Ser1*, which lacks a dihydrouridine (DHU) arm, resulting in a simple loop at this location (Appendix A), which is also common in insects [81,82,83]. The length of the *tRNA*-*Ser1* genes is 67 bp in all 21 *Chlorophorus* species, with a UCU codon in the anticodon loop. Additionally, all *tRNA*-*Lys* genes harbor a UUU anticodon, a feature unique to Chrysomeloidea and consistent with previous research [44,46]. Several unmatched base pairs, such as G–U and U–G, were observed in the tRNA stems of all 21 species and found on all four stems. Unmatched base pairs can be revised via editing processes or may symbolize abnormal matches [84].

The two rRNA genes are encoded on negative strands, and their locations and characteristics are consistent with those of previously studied Cerambycids [75,76,78,79]. The 16S *rRNA* gene was found located between *tRNA*-*Leu1* and *tRNA*-*Val*, while the 12S *rRNA* gene is located between *tRNA*-*Val* and the control region. The base length of 16S *rRNA* ranges from 1186 bp to 1269 bp, and that of 12S *rRNA* ranges from 752 bp to 770 bp.

#### 3.1.5. Substitution Saturation Tests and Nucleotide Heterogeneity

We present both ML and BI trees based on two datasets (13 PCGs + 2 rRNAs + 22 tRNAs, 13 PCGs + 2 rRNAs) for a total of 35 mitochondrial genomes and present BI trees use Phylobayes based on PCG123 and PCG12 datasets. The four datasets were analyzed for nucleotide substitution saturation. The results show that the ISS (simple index of substitution saturation) < ISS.c (critical ISS value) and *p* < 0.05 (Appendix A), indicating that the datasets were not saturated with substitutions. All these data types are thus suitable for use in phylogenetic analyses. We evaluated the heterogeneity in nucleotide divergence via pairwise comparisons in multiple sequence alignment. The results indicate the low heterogeneity among the four datasets (Appendix A).

### 3.2. Phylogenetic Analysis

In this study, we used both maximum likelihood (ML) and Bayesian inference (BI) methods, and used two datasets (13PCGs + 2rRNAs, 13PCGs + 2rRNAs + 22tRNAs) to generate four phylogenetic trees. Additionally, we utilized Phylobayes under the CAT-GTR model to analyze two datasets (PCG123, PCG12). The results of the BI and ML analyses based on the four datasets are illustrated in Figure 2 and Appendix A. Despite the variations in the datasets and methodologies employed, the trees constructed in this study have congruent topologies.

#### 3.2.1. Monophyly of Genus *Chlorophorus*

In all phylogenetic trees constructed from the various datasets and methods, *Chlorophorus* form a monophyletic group, with a very high support value (ML tree, bootstrap values, BS = 100; BI tree, posterior probabilities, PP = 1) (Figure 2 and Appendix A).

Our results indicate that the genus *Chlorophorus* was recovered as monophyletic. This is in contrast to recent phylogenetic research based on multiple genes showing that the genus *Chlorophorus* is completely polyphyletic, such as by Lee and Lee (two mitochondrial genes, *COI* and 16S *rRNA*, and four nuclear genes, *wingless*, *CAD*, 18S, and 28S *rRNA*, 3 species included) [39] and Zamoroka (three mitochondrial genes, *COI*, 12S, and 16S *rRNA*, and two nuclear genes, 18S and 28S *rRNA*, 15 species included) [1]. Importantly, our datasets share only four species with Zamoroka’s study. Although species overlap is limited, our taxon sampling still incorporates representatives within the genus, contributing to robust conclusions.

One of the notable species, *C. diadema*, was placed outside *Chlorophorus* in both previous studies [1,39] and designated as the type species for the newly established genus *Sparganophorus* by Zamoroka [1]. This genus was later synonymized with *Chlorophorus* (*Humeromaculatus*) by Lazarev [2]. However, *C. diadema* was placed firmly within the *Chlorophorus* clade in our results, challenging the validity of *Sparganophorus*. The morphological evidence further corroborates this finding. Zamoroka diagnosed *Sparganophorus* based on characteristics such as the forehead’s median carina, antennal segment formula (1 = 3 > 5 = 4), a wide prosternal process, and a long first metatarsomere [1]. However, these characteristics are insufficient to distinguish *Sparganophorus* from *Chlorophorus*. Gressitt [18] defined *Chlorophorus* using similar characteristics, specifically noting that “the third antennal segment is no longer than the scape, and the first tarsal segment is no longer than the subsequent segments.” Thus, the phylogenetic and morphological data together strongly suggest that *Sparganophorus* is not a valid genus. However, our results do not support synonymy with the subgenus *Humeromaculatus*.

The subgenus *Humeromaculatus* was originally established by Özdikmen with *Cerambyx figuratus* as its type species [3] and later elevated to genus level by Zamoroka [1]. Our study identifies *C. simillimus* (assigned to *Humeromaculatus* by Zamoroka) as a sister species to *C. fraternus*, supporting the rejection of *Humeromaculatus* as a separate genus. This finding aligns with Lazarev [2], who argued against elevating *Humeromaculatus* to genus status. Zamoroka’s diagnosis of *Humeromaculatus* emphasized characteristics such as a trapezoidal forehead, antennal formula (1 = 3 = 5 > 4), an oblong pronotum, and a narrow prosternal process. However, inconsistencies arise when examining species previously placed in subgenera *Viridiphorus* and *Humeromaculatus*. For instance, species like *C. muscosus*, *C. quinquefasciatus*, and *C. japonicus* display antennal segment lengths (e.g., the third antennal segment subequal to the scape) that challenge the stability of these diagnostic characteristics.

The contradictory findings between our study and those of Zamoroka [1] and Lee and Lee [39] may be attributed to differences in taxon sampling and dataset composition. While our study includes fewer overlapping species with Zamoroka’s, we incorporated key representatives, such as *C. diadema* and *C. simillimus.* The monophyly of *Chlorophorus* observed here, with strong support values, highlights the need for comprehensive taxon sampling and integrative approaches combining morphological and molecular evidence to resolve these discrepancies robustly.

#### 3.2.2. Subgeneric Classification

The remaining *Chlorophorus* species, apart from *C. furtivus*, *C. proannulatus*, and *C. siegriedae*, formed two clades. One clade displayed the topology of ((*C. arciferus* + (*C.* cf. *punctiger tamdaoensis* + *C. copiosus*)) + (*C. annularoides* + (*C. orbatus* + (*C. annularis* + (*C. insidiosus* + *C. annularis* NC 061058))))). The other clade comprised (*C. diadema diadema* + ((*C. tredecimmaculatus* + (*C. simillimus* + *C. fraternus*)) + (*C. douei* + (*C. quatuordecimmaculatus* + (*C. lingnanensis* + (*C. montanus* + (*C. annulatus* + (*C. diversicolor* + *C. coniperda*)))))))) (Figure 2).

Özdikmen [38] studied Turkish *Chlorophorus* species to propose a subgeneric arrangement with five subgenera and then proposed 36 subgenera for the world fauna based on elytral markings [3]. However, our phylogenetic results in this study did not support monophyly of the subgenera *Humeromaculatus* Özdikmen, 2011; *Immaculatoides* Özdikmen, 2022; *Brevenotatus* Özdikmen, 2022; and *Chlorophorus* Chevrolat, 1863 (Figure 2).

The subgenus *Humeromaculatus* includes 35 species/subspecies. In this study, *C.* (*Humeromaculatus*) *diadema diadema* was separated from *C.* (*Humeromaculatus*) *coniperda* and *C.* (*Humeromaculatus*) *diversicolor*. The subgenus *Brevenotatus* comprises 27 species/subspecies, of which four included in the present phylogenetic analysis were found to be completely polyphyletic. *C*. *copiosus*, *C*. cf. *punctiger tamdaoensis* were a sister group and clustered with *C. arciferus.* Meanwhile, *C. simillimus* clustered with *C. fraternus* and *C. tredecimmaculatus*.

The subgenus *Immaculatoides* is characterized by its upper side, which is predominantly unicolorous, often with varying degrees of dark spots. Notably, although *C. montanus* and *C. fraternus* exhibit similar elytral markings that comply with this diagnosis of *Immaculatoides*, our phylogenetic analysis reveals that these two species do not cluster together, indicating *Immaculatoides* is not monophyly.

*C. annulatus* and *C. douei* were separated from the species belonging to subgenus *Chlorophorus.* They clustered with *C. quatuordecimmaculatus*, *C. lingnanensis*, *C. montanus*, *C. coniperda*, and *C. diversicolor* in our study. This differs from Zamoroka’s research [1], which indicated that *C. annulatus* is closer to *C. annularis* (with SH-like value 0.69).

#### 3.2.3. The Limitations of Elytral Markings in Subgeneric Classification

The identification of the genus is primarily based on the elytral marking pattern [19,20]. Özdikmen’s work has established a fundamental framework of subgeneric classification. However, defining elytral marking patterns can be challenging in certain cases. For instance, the elytral markings of *C. simillimus* and *C. tredecimmaculatus* are highly similar (Appendix A), despite belonging to two different subgenera, *Brevenotatus* and *Sexnigromaculatus*, respectively. According to Özdikmen [38] the humeral spots of *Brevenotatus* are adjacent to the basal strip, whereas those of *Sexnigromaculatus* are not. However, based on our observations, the humeral spots of both species do not appear to be adjacent to the basal strip.

Furthermore, variations in elytral marking patterns within species of this genus have been documented [85], and we have observed a similar phenomenon. For instance, the typical elytral marking of *C. tredecimmaculatus* generally includes three spots on each elytron, excluding the humeral spot; this aligns with the definition of the subgenus *Sexnigromaculatus*. However, several specimens exhibit only two spots on each elytron (Appendix A).

Although *C. montanus* and *C. fraternus* exhibit similar elytral markings, phylogenetic analyses indicate that they belong to different clades. Additionally, the distinct shapes of the tegmen and the different sclerites of the endophallus suggest that these species are not closely related. Therefore, the similarity of elytral markings alone should not be considered a reliable indicator of close evolutionary relationships.

#### 3.2.4. Implication of the Sclerites of the Endophallus for Taxonomy of the Genus

Male genitalia typically exhibit stable specific differences, making them important for species identification. The taxonomic significance of male genitalia, including the length of the paramere, the length ratio of the struts of the median lobe, and the presence of tiny spines on the endophallus, has been demonstrated in previous studies [86,87]. Morphological characteristics of male genitalia are frequently employed in phylogenetic analyses within the family Cerambycidae [88].

Our findings support the hypothesis of a close evolutionary relationship among certain species, as indicated by both phylogenetic analyses and similarities in their male genitalia. Notably, the sclerites of the endophallus may provide significant phylogenetic signals (Figure 2). For instance, the clade comprising *C. fraternus*, *C. simillimus*, and *C. tredecimmaculatus* exhibits similarities in male genitalia, characterized by stouter parameres and two pairs of endophallus sclerites: one pair is “/\”-shaped, while the other pair is smaller.

The clade of the subgenus *Chlorophorus*, with the exception of *C. annulatus* and *C. douei*, shares similarities in the sclerites of the endophallus, which possess a hook. Although *C. annulatus* is classified under the subgenus *Chlorophorus*, its endophallus sclerites are similar to those of *C. coniperda* and *C. diversicolor*. Furthermore, *C. diadema diadema*, belonging to the subgenus *Humeromaculatus*, exhibits a distinctly different shape of endophallus sclerites compared to *C. coniperda* and *C. diversicolor*.

*C. copiosus*, *C.* cf. *punctiger tamdaoensis*, and *C. arciferus* together form a clade, wherein their endophallus sclerites are separate from one another, rather than interconnected. Although the sclerites of the endophallus have been documented in Cerambycidae [89,90], their phylogenetic significance has yet to be evaluated. And unfortunately, a significant number of species within *Chlorophorus* still lack comprehensive descriptions of male genitalia. Thus, conducting comparative morphological studies of male genitalia of the genus is essential.

#### 3.2.5. The Identification of *C. annularis*

*C. annularis* (accession no. NC 061058) does not cluster with *C. annularis* sequenced in this study. We constructed a neighbor-joining (NJ) tree to test for misidentification and calculated the pairwise distances based on the *COI* (658 bp) from NCBI and the mitogenome of *Chlorophorus* sequenced in this study using MEGA v.11 [61]. *C. annularis* (NC 061058) is not placed within the *C. annularis* clade (Appendix A). The genetic distance values for NC 061058 and the remaining *C. annularis* samples ranged from 0.104 to 0.111, while the genetic distance values within *C. annularis* ranged from 0.002 to 0.018 (Appendix A). Therefore, the sample of NC 061058 might have been misidentified as *C. annularis*.

## 4. Conclusions

We sequenced, assembled, and annotated the mitochondrial genomes of 21 *Chlorophorus* species in this study. The lengths of the *Chlorophorus* mitochondrial genomes were found to range from 15,387 bp to 15,779 bp, exhibiting no rearrangements or deletions, and the sequences and compositions are extremely conserved across all genes. The three most commonly used amino acids in *Chlorophorus* are Ile, Phe, and Leu1. The analyses of non-synonymous to synonymous substitution ratios showed that *ATP8* has the highest evolutionary rate, while *COI* has the lowest. Phylogenetic trees were constructed using the BI and ML methods based on four different datasets. The topology determined from all analyses supports the monophyly of the genus *Chlorophorus*. However, the analyses did not provide support for the monophyly of the subgenera *Chlorophorus* (*Humeromaculatus*) Özdikmen, 2011; *Chlorophorus* (*Immaculatoides*) Özdikmen, 2022; *Chlorophorus* (*Brevenotatus*) Özdikmen, 2022; and *Chlorophorus* (s. str.) Chevrolat, 1863.

The classification of the genus is notably complex. As pointed out by Özdikmen, the subgeneric arrangement of *Chlorophorus* is far from being conclusively resolved [3]. In this study, we refrained from making any changes to the subgeneric classification due to limited taxon sampling. Although this study does not provide a definitive resolution to the classification system of the genus, we anticipate that future comparative morphological studies of male genitalia will be conducted, along with more extensive taxon sampling and the analysis of additional molecular markers.

## Figures and Tables

**Figure 1 insects-16-00008-f001:**
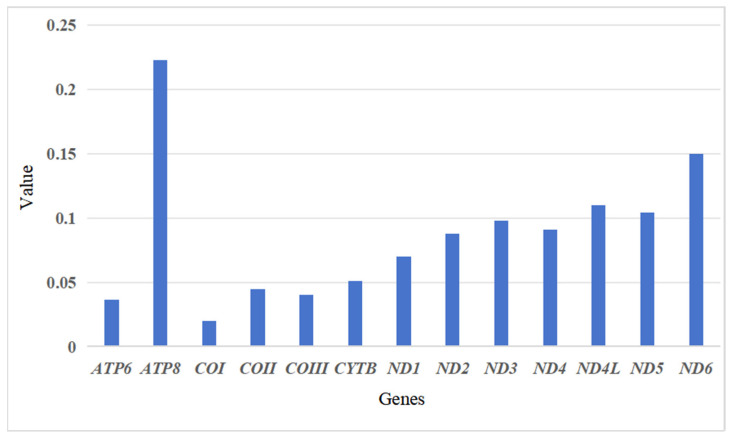
Evolutionary rates of genes encoding proteins in the *Chlorophorus* mitochondrial genomes.

**Figure 2 insects-16-00008-f002:**
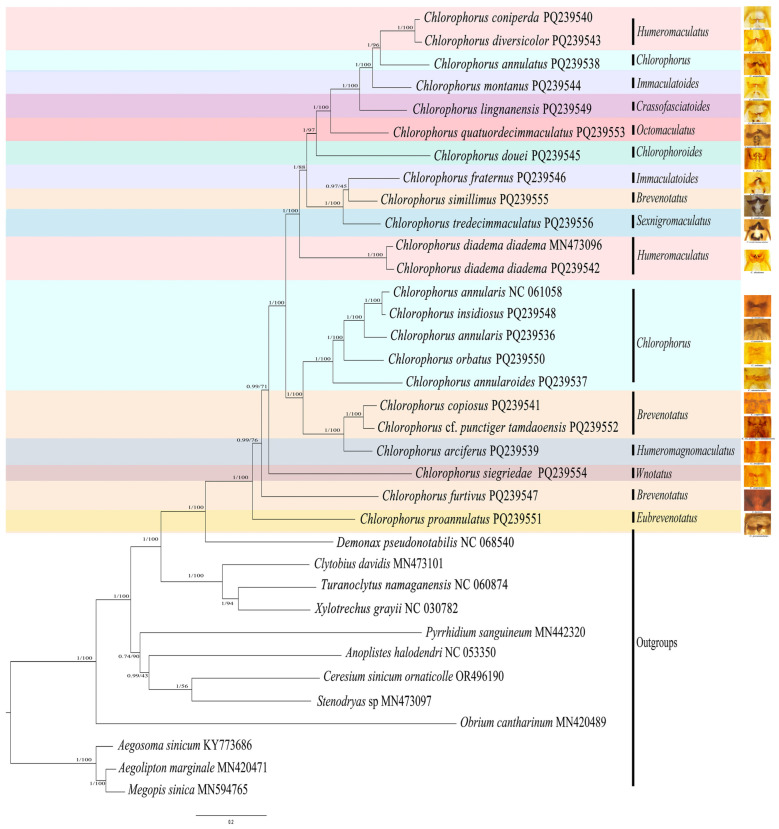
Phylogenetic tree of *Chlorophorus* based on the PCGs + rRNAs + tRNAs datasets obtained using MrBayes and IQtree. The numbers on branches are Bayesian posterior probabilities (PP, (**left**)) and bootstrap values (BS, (**right**)). The sclerites of endophallus are given.

**Table 1 insects-16-00008-t001:** Information about samples used in this study and their NCBI GenBank accession numbers.

Subfamily	Subgenus of *Chlorophorus*	Species	Sample Locality	GenBank Accession	Reference
Cerambycinae	*Brevenotatus*	*Chlorophorus copiosus*	Yunnan	PQ239541	This study
Cerambycinae	*Brevenotatus*	*Chlorophorus furtivus*	Yunnan	PQ239547	This study
Cerambycinae	*Brevenotatus*	*Chlorophorus* cf. *punctiger tamdaoensis*	Guangxi	PQ239552	This study
Cerambycinae	*Brevenotatus*	*Chlorophorus simillimus*	Heilongjiang	PQ239555	This study
Cerambycinae	*Chlorophoroides*	*Chlorophorus douei*	Yunnan	PQ239545	This study
Cerambycinae	*Chlorophorus*	*Chlorophorus annularis*	Unknown	NC 061058	Unpublished
Cerambycinae	*Chlorophorus*	*Chlorophorus annularis*	Yunnan	PQ239536	This study
Cerambycinae	*Chlorophorus*	*Chlorophorus annularoides*	Yunnan	PQ239537	This study
Cerambycinae	*Chlorophorus*	*Chlorophorus annulatus*	Guizhou	PQ239538	This study
Cerambycinae	*Chlorophorus*	*Chlorophorus insidiosus*	Yunnan	PQ239548	This study
Cerambycinae	*Chlorophorus*	*Chlorophorus orbatus*	Yunnan	PQ239550	This study
Cerambycinae	*Crassofasciatoides*	*Chlorophorus lingnanensis*	Guizhou	PQ239549	This study
Cerambycinae	*Eubrevenotatus*	*Chlorophorus proannulatus*	Yunnan	PQ239551	This study
Cerambycinae	*Humeromaculatus*	*Chlorophorus coniperda*	Yunnan	PQ239540	This study
Cerambycinae	*Humeromaculatus*	*Chlorophorus diadema diadema*	Beijing	MN473096	[51]
Cerambycinae	*Humeromaculatus*	*Chlorophorus diadema diadema*	Gansu	PQ239542	This study
Cerambycinae	*Humeromaculatus*	*Chlorophorus diversicolor*	Yunnan	PQ239543	This study
Cerambycinae	*Humeromagnomaculatus*	*Chlorophorus arciferus*	Xizang	PQ239539	This study
Cerambycinae	*Immaculatoides*	*Chlorophorus fraternus*	Guizhou	PQ239546	This study
Cerambycinae	*Immaculatoides*	*Chlorophorus montanus*	Yunnan	PQ239544	This study
Cerambycinae	*Octomaculatus*	*Chlorophorus quatuordecimmaculatus*	Yunnan	PQ239553	This study
Cerambycinae	*Sexnigromaculatus*	*Chlorophorus tredecimmaculatus*	Yunnan	PQ239556	This study
Cerambycinae	*Wnotatus*	*Chlorophorus siegriedae*	Yunnan	PQ239554	This study
Cerambycinae	Outgroup	*Demonax pseudonotabilis*	Sichuan	NC 068540	[52]
Cerambycinae	Outgroup	*Clytobius davidis*	Beijing	MN473101	[51]
Cerambycinae	Outgroup	*Turanoclytus namaganensis*	Unknown	NC 060874	Unpublished
Cerambycinae	Outgroup	*Xylotrechus grayii*	Unknown	NC 030782	Unpublished
Cerambycinae	Outgroup	*Pyrrhidium sanguineum*	Czech Republic	MN442320	[51]
Cerambycinae	Outgroup	*Anoplistes halodendri*	Unknown	NC 053350	Unpublished
Cerambycinae	Outgroup	*Ceresium sinicum ornaticolle*	Unknown	OR496190	Unpublished
Cerambycinae	Outgroup	*Stenodryas* sp.	Beijing	MN473097	[51]
Cerambycinae	Outgroup	*Obrium cantharinum*	Czech Republic	MN420489	[51]
Prioninae	Outgroup	*Aegosoma sinicum*	Unknown	KY773686	Unpublished
Prioninae	Outgroup	*Aegolipton marginale*	Yunnan	MN420471	[51]
Prioninae	Outgroup	*Megopis sinica*	Unknown	MN594765	[53]

Note: For the subgeneric classification of *Chlorophorus*, refer to Özdikmen [3].

## Data Availability

The new mitogenome sequences have been deposited in GenBank of NCBI under the accession numbers PQ239536–PQ239556.

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
