# Peer review of "Comparative Mitogenomics in the Genus Chlorophorus (Coleoptera: Cerambycidae) and Its Phylogenetic Implications"

_insects, 2024, doi:10.3390/insects16010008_

Round 1

Reviewer 1 Report

Comments and Suggestions for Authors

This study explores the phylogenetic relationships and subgeneric classification of the species-rich genus Chlorophorus using mitochondrial genomic data from 21 species. Through high-throughput sequencing, the authors reconstructed the phylogeny using maximum likelihood (ML) and Bayesian inference (BI) methods. The findings revealed conserved mitochondrial genome arrangements and AT-biased composition across all species.

The major issue of this paper is that the phylogenetic methods used is not well justified. The software implemented in PhyloSuite are IQTree and MrBayes, and both do not consider among-site compositional heterogeneity in the data. This work definitely contains new data, but the phylogenetic analyses appear to be out-of-data and require reconsideration

Mitogenome compositional heterogeneity:

Why were the nucleotide dataset analyzed only under a site-homogeneous model? The PhyloBayes software, which allows different substitution processes for amino acid replacement at various sites, produced a tree that best matched known higher-level taxa and defined basal relationships in Hemiptera and Coleoptera (see Timmermans et al. 2015). Timmermans et al. (2015) also showed that the compositional heterogeneity cannot be eliminated for some mitochondrial genes, but dense taxon sampling and the use of appropriate Bayesian analyses (CAT-GTR+G model) can still produce robust phylogenetic trees. To infer a phylogenetic tree based on mitogenomic data, it is always better to use the CAT-GTR model accounting for compositional heterogeneity, as this phenomenon has been observed not only in Coleoptera (Cai et al., 2020), but also Neuroptera and Hemiptera. As such, the authors need to do extra runs of their data sets (cd12 and cd123) using the CAT-GTR model in PhyloBayes.   

Data availability: The inclusion of a detailed description of the molecular dataset and the final matrix will enhance the transparency and reproducibility of the study. The authors should make available the finial supermatrix and the unaligned, aligned, and add the link to the data in the methodology section.

Author Response

Dear reviewer,

     We sincerely appreciate for your valuable suggestions and comments, particularly regarding the phylogenetic methods employed and the data presented in our study. Your insights will undoubtedly enhance the clarity and strength of our manuscript.

Comment 1. The major issue of this paper is that the phylogenetic methods used is not well justified. The software implemented in PhyloSuite are IQTree and MrBayes, and both do not consider among-site compositional heterogeneity in the data. As such, the authors need to do extra runs of their data sets (cd12 and cd123) using the CAT-GTR model in PhyloBayes.

Response 1: We acknowledge the limitations of using models that do not account for among-site compositional heterogeneity. To address this, we performed additional analyses using Phylobayes with the CAT+GTR model on two datasets (cd12 and cd123). These analyses showed the same topologies as those generated by the maximum likelihood (ML) and Bayesian inference (BI) methods previously applied. This consistency in results suggests that our findings remain robust despite the differing methodologies.

Comment 2: This work definitely contains new data, but the phylogenetic analyses appear to be out-of-data and require reconsideration .

Response 2:  We greatly appreciate your suggestion regarding the limitations of the data. We attempted to expand our datasets but, unfortunately, we have not been able to obtain additional material at this time. We acknowledge that the limited taxon sampling restricts our ability to draw comprehensive conclusions about phylogeny, which is why we have chosen not to make any significant alterations to the subgeneric classification in this study.

While we understand that our study may not fully resolve the classification system of the genus, we believe it raises critical questions about the polyphyletic nature of certain subgenera. We stress that future comparative morphological studies—particularly focusing on male genitalia—combined with more extensive taxon sampling and the incorporation of additional molecular markers, will be crucial in addressing these challenges and refining our understanding of the phylogenetic relationships within Chlorophorus.

Comment 3. Data availability: The inclusion of a detailed description of the molecular dataset and the final matrix will enhance the transparency and reproducibility of the study. The authors should make available the finial supermatrix and the unaligned, aligned, and add the link to the data in the methodology section.

Response 3: We appreciate your suggestion regarding data transparency and reproducibility. In response, we have included the final supermatrix, as well as the unaligned and aligned matrices, in the supplementary materials of the manuscript.

Once again, thank you for your constructive comments and suggestions, which have greatly improved our manuscript. We hope that our revisions meet your approval.

With our best regards!

Sincerely yours,

Zhu LI

Reviewer 2 Report

Comments and Suggestions for Authors

In this study, Fu et al. assemble and annotate the mitochondrial genomes of 21 species of the genus Chlorophorus. They perform a comparative analysis of the composition and organization of mtDNA sequences among species and use this data to conduct a phylogenetic study of the genus, including other species from the subfamily Clytini as outgroups.

In my opinion, the study is well-conducted and provides a significant amount of new information on species of the genus Chlorophorus, warranting publication.

However, certain points need clarification, especially the discrepancies with previous studies.

The most notable result is that the mitogenome analysis supports the monophyly of this genus, in contrast to prior studies using mitochondrial and nuclear genes. The authors note that previous studies clearly demonstrated that the genus Chlorophorus is polyphyletic (Zamoroka 2021). Are the species included in this study different from those used by Zamoroka? Could this justify the contradictory results? This aspect should be addressed in greater depth by the authors, as it represents the most intriguing finding of the study.

Additionally, I have a few minor suggestions:

Materials and Methods: Please specify the method used for DNA extractions.

Line 173: Which species has the largest non-coding region?

Lines 203–213: The use of incomplete stop codons (T-- or TA-) is common in insects. For readers less familiar with this phenomenon, it would be helpful to explain how the stop codon is completed through poly-A tail addition.

Lines 203–213: Ile, Phe, and Leu1 are also the most common amino acids in many other insect species. Please include references to support this observation.

Lines 215–224: The absence of the DHU arm in tRNA structures is also common in insects. Please comment on this and include references.

Line 241: There appears to be a typographical error, as it refers to Figure 6, but the correct reference is Figure 2.

Author Response

Dear Reviewer,

We greatly appreciate your comments and suggestions, particularly regarding the details of our methodology and scientific writing. Your feedback has been invaluable in enhancing the clarity and depth of our manuscript.

Comment 1: The most notable result is that the mitogenome analysis supports the monophyly of this genus, in contrast to prior studies using mitochondrial and nuclear genes. The authors note that previous studies clearly demonstrated that the genus Chlorophorus is polyphyletic (Zamoroka 2021). Are the species included in this study different from those used by Zamoroka? Could this justify the contradictory results? This aspect should be addressed in greater depth by the authors, as it represents the most intriguing finding of the study.

Response 1: Thank you for highlighting this important aspect of our study. We have revised this section to provide a more in-depth discussion of the contrasting findings between our research and those of Zamoroka and Lee and Lee.

We believe that differences in taxon sampling and dataset composition are significant factors contributing to these discrepancies. Our study includes four species that overlap with Zamoroka's research: C. annulatus, C. annularis, C. diadema, and C. simillimus. C. annularis is the type species of the genus Chlorophorus, and C. annulatus represents the subgenus Chlorophorus.

Interestingly, C. diadema was previously placed outside the genus Chlorophorus in both previous studies based on multiple markers and was designated as the type species for the newly established genus Sparganophorus by Zamoroka. Our findings challenge the validity of Sparganophorus, suggesting that it may not be a separate lineage as previously thought. Furthermore, C. simillimus was assigned to the subgenus Humeromaculatus by Zamoroka, and our results reject the validity of this subgenus as well. Our results provide strong support for the monophyly of Chlorophorus across various datasets and methodologies, by high support values. The polyphyly suggested by Zamoroka lacks sufficient morphological evidence and the genus he proposed has been synonymized by Lazarev.

This finding highlights the importance of comprehensive taxon sampling and underscores the need for integrative approaches that combine molecular and morphological data to resolve phylogenetic relationships within this complex genus.

Comment 2: Materials and Methods: Please specify the method used for DNA extractions.

Response 2: We have specified the method for DNA extraction. (Line 113: Genomic DNA was extracted from an adult’s muscle tissue of the prothorax and legs by the modified CTAB method [51]. )

Comment 3: Line 173: Which species has the largest non-coding region?

Response 3: We have added the specific species in our manuscript. (Line 188: The non-coding regions range from 813 bp (C. diadema diadema) to 1205 bp (C. douei) in length.)

Comment 4: Lines 203–213: The use of incomplete stop codons (T-- or TA-) is common in insects. For readers less familiar with this phenomenon, it would be helpful to explain how the stop codon is completed through poly-A tail addition.

Response 4: We have explained it. (Lines 223–225: All 13 PCGs use TAA/TAG or a single T as stop codons, and the T is replenished by post-transcriptional polyadenylation at the 3’ end [77], consistent with previous studies on Cerambycidae [75, 76, 78, 79]. )

Comment 5: Lines 203–213: Ile, Phe, and Leu1 are also the most common amino acids in many other insect species. Please include references to support this observation.

Response 5: We have added some references to support the observation. (Line 228: The statistics of the relative synonymous codon usage showed that the most commonly used amino acids in the Chlorophorus mitochondrial genome are Ile, Phe, and Leu1, which are also the most common amino acids in many other insect species [80–83].)

Comment 6: Lines 215–224: The absence of the DHU arm in tRNA structures is also common in insects. Please comment on this and include references.

Response 6: We have included references to support this observation, highlighting that this phenomenon is indeed common in insects. (Line 238: ... it lacks a dihydrouridine (DHU) arm, resulting in a simple loop at this location (Figures S43–S63), which is also common in insects [81–83].)

Comment 7 :Line 241: There appears to be a typographical error, as it refers to Figure 6, but the correct reference is Figure 2.

Response 7: We have the reference to Figure 2.

Once again, thank you for your constructive comments and suggestions, which have greatly improved our manuscript. We hope that our revisions meet your approval.

With our best regards!

Sincerely yours,

Zhu LI